# Mechanically Tunable, Compostable, Healable and Scalable Engineered Living Materials

Avinash Manjula-Basavanna [1,2,3] ✉, Anna M. Duraj-Thatte [2] & Neel S. Joshi [1] ✉

Advanced design strategies are essential to realize the full potential of engineered living materials, including their biodegradability, manufacturability, sustainability, and ability to tailor functional properties. Toward these goals, we present mechanically engineered living material with compostability, healability, and scalability – a material that integrates these features in the form of a stretchable plastic that is simultaneously flushable, compostable, and exhibits the characteristics of paper. This plastic/paper-like material is produced in scalable quantities ($0.5–1\,g\,L^{-1}$), directly from cultured bacterial biomass (40%) containing engineered curli protein nanofibers. The elongation at break (1–160%) and Young's modulus (6-450 MPa) is tuned to more than two orders of magnitude. By genetically encoded covalent crosslinking of curli nanofibers, we increase the Young's modulus by two times. The designed engineered living materials biodegrade completely in 15–75 days, while its mechanical properties are comparable to petrochemical plastics and thus may find use as compostable materials for primary packaging.

The emerging field of Engineered Living Materials (ELMs) employs synthetic biology design principles to harness the programmability and the manufacturing capabilities of living cells to produce functional materials[1–4]. ELMs research not only provides avenues to integrate life-like properties into materials but also aims to realize de novo functionalities that are not found in natural or synthetic materials[5–21]. In recent years, several ELMs have been developed to demonstrate various functionalities such as adhesion, catalysis, mineralization, remediation, wound healing, and therapeutics etc[22–31]. ELMs that are mechanically stiff or soft have also been reported, but the rational modulation of mechanical properties to a wide range through genetic programming remains elusive[5,6,9–11,25,32]. In this regard, we present an ELM called MECHS, which stands for Mechanically Engineered Living Material with Compostability, Healability, and Scalability (Fig. 1).

Advances in biomanufacturing are important at a time when human-made materials have been estimated to outweigh all the living biomass of planet Earth[33]. The existing paradigm of a linear materials economy (make-use-dispose) for synthetic materials is causing potentially irreversible damage to our ecosystem in the form of pollution and global warming. While many strategies will need to be employed to address these challenges, it is clear that bio-based manufacturing will need to be part of the solution[34]. Inspired by natural systems that utilize sustainable feedstocks and energy-efficient processes, coupled with their biodegradation to initiate a new cycle, biomanufacturing should strive to create materials that have similar recyclability or potential for conversion to benign components to create a circular material economy[35–37]. Such nature-inspired sustainable solutions enabled by biomanufacturing will also make inroads toward practical implementation through a combination of appropriate government policies, public interest, and investment[38].

Previously, we had reported a bioplastic known as AquaPlastic composed of recombinant protein nanofibers produced by *E. coli*[9]. It

[1]Department of Chemistry and Chemical Biology, Northeastern University, Boston, Massachusetts, USA. [2]Department of Biological Systems Engineering, Virginia Polytechnic Institute and State University, Blacksburg, Virginia, USA. [3]Department of Bioengineering, Northeastern University, Boston, Massachusetts, USA. ✉e-mail: mbavinash@northeastern.edu; ne.joshi@northeastern.edu

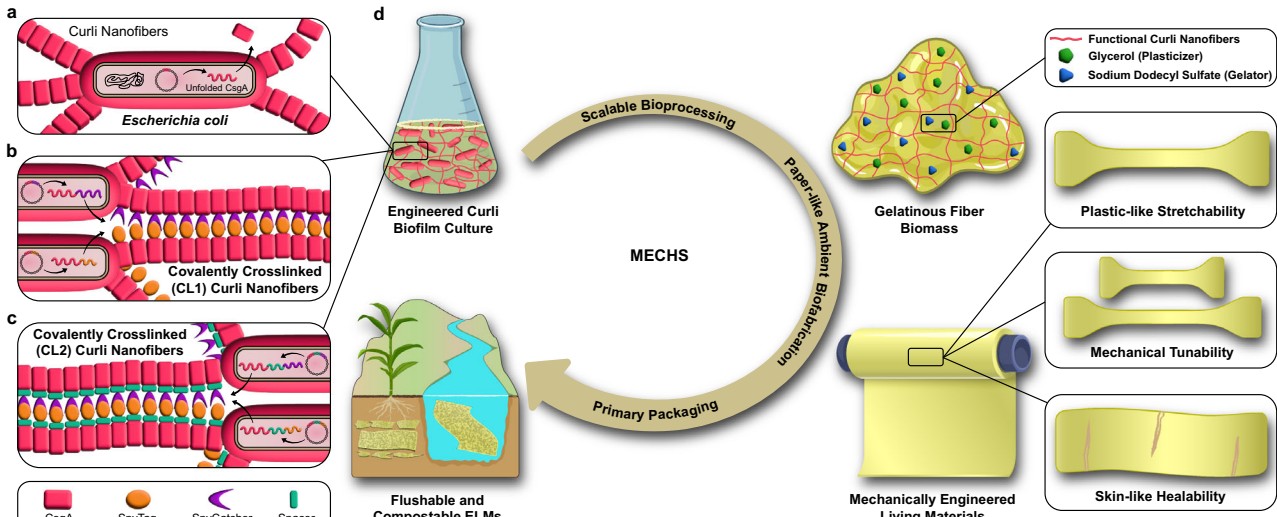

**Fig. 1 | Schematic summary of Mechanically Engineered Living Materials with Compostability, Healability and Scalability (MECHS). a** Native and (**b**, **c**) functional curli nanofibers were separately produced from engineered *Escherichia coli*. **d** The treated biomass of engineered *E. coli* was dried ambiently to biofabricate MECHS films in a scalable manner. MECHS films exhibit plastic-like stretchability, mechanical tunability, and skin-like healability. Parts of the schematics were created in BioRender. Duraj-thatte, A. (2024), BioRender.com/x16s696, BioRender.com/u23i785 and BioRender.com/p06e527.

exhibited Young's modulus of ~1 GPa and ultimate tensile strength of ~25 MPa, comparable to petrochemical plastics and other bioplastics[9]. AquaPlastic was also resistant to various chemicals (e.g., acid, base, and organic solvents), and could adhere to and coat a wide range of surfaces, protecting them from wear and environmental conditions[9]. However, the broad utility of AquaPlastic was limited due to its brittleness and lack of scalability. In addition, we had earlier shown that whole microbial biomass could be dried to form cohesive and glassy stiff materials with a streamlined fabrication and higher yields compared to AquaPlastic, at the expense of tunability[12].

In this work, we report a fabrication strategy to combine whole cellular biomass and engineered extracellular matrix protein nanofibers that facilitate tuning of their mechanical properties. Our material, MECHS, exhibits properties similar to both plastic and paper, showcasing: (1) a fabrication strategy that enables large-scale production of flexible films at ambient conditions, analogous to paper manufacturing; (2) genetic engineering to tailor their tensile stiffness and strength; (3) compositional and morphological analysis; (4) compostability, (5) a landscape of achievable mechanical properties comparable to conventional petrochemical plastics, bioplastics and other relevant bio- and synthetic materials; and, (6) prototypes for disposable packaging applications, contributing to the creation of a sustainable circular material economy.

## Results

### Biofabrication of MECHS

MECHS is fabricated from a combination of whole *E. coli* cells and engineered recombinant curli nanofibers. Curli are an extracellular matrix component of microbial biofilms and are composed of nanofibers self-assembled from a protein building block, CsgA (Fig. 1a–d, Table 1 and Supplementary Table 1)[39]. Curli nanofibers are resistant to heat, solvents, pH, detergents, and denaturants, and thus serve as a good biopolymeric scaffold for robust materials[40]. To express the recombinant curli nanofibers, we used an *E. coli* strain that we previously developed (PQN4), in which the chromosomal curli genes (*csgBAC*, *csgDEFG*) have been deleted[41]. PQN4 was transformed with a pET21d plasmid vector encoding a synthetic curli operon, *csgBACEFG*, containing all the genes necessary for CsgA production, secretion, and extracellular assembly. In a typical biofabrication of MECHS, the curli-containing *E. coli* biomass was treated with 1–5% (w v⁻¹) of sodium dodecyl sulfate (SDS) to obtain a gelatinous substance, which enables facile casting in a silicone mold. Ambient drying in the mold resulted in films that were brittle and, in some cases, (1% and 2% SDS) convoluted (Supplementary Figs. 1, 2). To achieve flexible MECHS films, we added glycerol (1–5% w v⁻¹), a plasticizer commonly used with bioplastics, to the gelatinous curli biomass prior to casting (Supplementary Figs. 3, 4)[42].

MECHS films that had been pre-treated only with SDS (i.e., gelator) and no glycerol (i.e., plasticizer), were brittle as measured by tensile mechanical tests, with elongation at break values of $0.6 \pm 0.4\%$ (Fig. 2a–e and Supplementary Figs. 5a and 6). With 1% plasticizer, the elongation at break was found to increase considerably to $10.2 \pm 6.9\%$ (Fig. 2b–f and Supplementary Fig. 5b). Similarly, as the plasticizer content increased to 2%, 3%, 4%, and 5%, the elongation at break increased significantly to $35.5 \pm 7.7\%$, $70.1 \pm 16.3\%$, $101.9 \pm 28.8\%$, and $159.3 \pm 25\%$ respectively (Fig. 2b–e, g–**j** and Supplementary Fig. 5c–f). On the other hand, the corresponding Young's modulus decreased from $450 \pm 206.4$ MPa to $6.6 \pm 1.7$ MPa as the plasticizer amount increased (Fig. 2d). Ultimate tensile strength values of MECHS films also decreased with increasing plasticizer (Fig. 2e). Overall, our method further streamlines the fabrication of flexible MECHS films from our previous demonstrations by casting directly from whole microbial biomass, without the need for filtration and extensive washing[9]. However, it also provides an opportunity to tailor their mechanical properties by two orders of magnitude by the inclusion of the engineered curli nanofibers and a plasticizer.

### Genetically engineered curli nanofibers to tailor the mechanical properties

Motivated by the above results, we genetically engineered the curli nanofibers to further modulate the mechanical properties of MECHS. We previously developed a Biofilm Integrated Nanofiber Display (BIND), wherein genetic fusions to CsgA are used to modulate material properties of assembled curli nanofibers[41]. During extracellular self-assembly, the robust β-helical blocks of CsgA fusions, stack on top of each other to form functional curli nanofibers with the desired peptide/protein fusions displayed on their surface. We used the genetic programmability of BIND to increase the stiffness of MECHS through covalent crosslinking. To achieve this, we utilized the third generation of split proteins derived from the adhesion domain, CnaB2 of

**Table 1 | Amino acid sequences of peptide/protein domains of MECHS variants**

| Peptide / Protein | Amino Acid Sequence | Length |
|---|---|---|
| CsgA | GVVPQYGGGGNHGGGGNNSGPNSELNIYQYGGGNSALALQTDARNSDLTITQHGGGNGADVGQGSD DSSIDLTQRGFGNSATLDQWNGKNSEMTVKQFGGGNGAAVDQTASNSSVNVTQVGFGNNATAHQY | 131 |
| Linker | GGSGSSGSGGSGGGSGSSGSGGSGGGSGSSGSGGSG | 36 |
| SpyTag | RGVPHIVMVDAYKRYK | 16 |
| SpyCatcher | VTTLSGLSGEQGPSGDMTTEEDSATHIKFSKRDEDGRELAGATMELRDSSGKTISTWISDGHVKDFYLY PGKYTFVETAAPDGYEVATPIEFTVNEDGQVTVDGEATEGDAHT | 113 |
| Spacer | KVLILACLVALALARETIESLSSSEESITEYKQKVEKVKHEDQQQGEDEHQDKIYPSFQPQPLIYPFVEPIP YGFLPQNILPLAQPAVVLPVPQPEIMEVPKAKDTVYTKGRVMPVLKSPTIPFFDPQIPKLTDLENLHLPLPL LQPLMQQVPQPIPQTLALPPQPLWSVPQPKVLPIPQQVVPYPQRAVPVQALLLNQELLLNPTHQIYPV TQPLAPVHNPISV | 225 |

*Streptococcus pyogenes* (SpyTag/SpyCatcher), wherein a spontaneous reaction between the side chains of lysine and aspartic acid residues results in the formation of an isopeptide bond[43]. This amide bond formation was reported to have high reactivity with > 90% completion in 15 min at 10 nM concentration, and for 10 µM, the half-time was less than 30 s[43]. Moreover, the reaction does not require any activating groups and is highly specific even in various complex biological media. SpyTag and SpyCatcher were each genetically grafted to CsgA via a linker to obtain CsgA-SpyTag and CsgA-SpyCatcher (Fig. 3a)[43]. These two CsgA constructs were expressed from separate plasmids in a co-culture, and the resulting curli biomass was used to fabricate MECHS films (denoted as CL1, Fig. 1b). The tensile tests of CL1 showed that their Young's modulus (51.6 ± 18.4 MPa) and ultimate tensile strengths (1.6 ± 0.4 MPa) were twice that of CsgA only (i.e., not crosslinked) based MECHS films, (Fig. 3c, d, f and Supplementary Figs. 7a, 8a). However, the elongation at break of CL1 was found to reduce to 29.8 ± 8.6% (Fig. 3e). We also tried analogous experiments with a large spacer (disordered protein domain of 225 amino acids) in between CsgA and the SpyTag/SpyCatcher domains (Figs. 3b, 1c)[44]. We introduced the large spacer for two reasons. 1) To verify if the observed increase in stiffness of CL1 was due to the covalent crosslinking of SpyTag and SpyCatcher. 2) To test if an intrinsically disordered large protein can modulate the mechanical properties such as stiffness, toughness, and elongation at break. MECHS films with this composition (i.e., CL2) were also found to have Young's modulus (46.6 ± 27.9 MPa), ultimate tensile strength (1.4 ± 0.7 MPa), and elongation at break (21.9 ± 6%), in the same range as that of CL1 (Fig. 3c–e and Supplementary Figs. 7b, 8a, b). The inter-fibrillar interactions of curli nanofibers in CsgA are that of relatively weaker supramolecular interactions, whereas, for CL1 and CL2, the inter-fibrillar covalent crosslinking of curli nanofibers is expected to resist the deformation of MECHS films leading to increased Young's modulus and ultimate tensile strength. However, this was achieved at the expense of elongation at break for CL1 and CL2 films. Although the covalent crosslinks enhance the stiffness of CL1 and CL2, we speculate that the softer biomass in the interstices between curli aggregates provides alternate pathways for crack propagation. Moreover, the slight decrease in Young's modulus and the ultimate tensile strength of CL2 in comparison to CL1 might be attributed to the effect of the disordered spacer domain. We reason that an even bigger spacer domain might lead to significant reductions in stiffness and enhanced extensibility.

## Composition and morphological analysis

Given the highly heterogeneous nature of the whole biomass that forms MECHS, we wanted to perform a detailed compositional analysis to understand the effects of various components therein. We focused on determining the amounts of curli biomass, gelator, and plasticizer in the final product, which may not be obvious from the fabrication protocol of MECHS. For example, treatment of the wet biomass with 1–5% gelator and/or plasticizer does not mean that the final MECHS film contains 1–5% gelator and/or plasticizer by mass since only a portion of

the original SDS and glycerol will associate with the cell pellet and the rest will be discarded with the supernatant, prior to film casting.

We first focused on estimating the amount of curli nanofibers present in the films on a per-weight basis using a standard Congo Red pull-down assay for curli quantification (Fig. 3g and Supplementary Fig. 9a). These relative amounts of curli were converted to absolute mass estimates with a calibration curve generated using purified curli nanofibers (wet weights of CsgA fused with His-tag i.e., CsgA-His). We estimated that 500 ml cultures of CsgA, CL1, and CL2 produced 530 ± 188 mg, 431 ± 159 mg, and 399 ± 154 mg of curli nanofibers, respectively (Fig. 3h and Supplementary Fig. 9b). The wet weights of whole cell pellets obtained from 500 ml cultures of CsgA, CL1 and CL2 were found to be 2647 ± 130 mg, 2483 ± 157 mg, 2490 ± 118 mg, respectively (Fig. 3g). Thus, we could estimate the percent of wet weight contributed by curli nanofibers for each construct (Fig. 3h). Notably, it is possible that the fused SpyTag/SpyCatcher domains may interfere with Congo Red binding, leading to an underestimation of curli nanofiber yields. On the other hand, 500 ml cultures of PQN4 with a sham plasmid (no curli operon) were found to have a wet cell pellet weight of 1936 ± 123 mg (Fig. 3g). It is interesting to note that the differences in wet pellet mass between curli-producing and sham plasmids roughly corresponds to the mass of curli nanofibers in each culture, calculated from the calibrated Congo Red binding assay (Fig. 3g, h).

We then set out for an extensive weight analysis to better understand the composition and the effect of various steps involved in the fabrication of MECHS. First, we determined that the ambient drying of the wet pellet of curli biofilm (without the treatment of gelator and plasticizer) results in a dry pellet with a weight percentage (dry to wet pellet) of 20.3 ± 1.8% (Supplementary Fig. 10a, b). The dry weight of MECHS films obtained after treatment of 1–5% of gelator was found to be about 100 mg, while the dry weight of the supernatant (collected from all the SDS treatment and water washings of cell pellets) was found to increase linearly (Supplementary Fig. 11a–d). It is to be noted that the experimentally obtained sum of weights of MECHS and the corresponding dry supernatant were consistent with their theoretically calculated weights (Supplementary Fig. 12a–e). Further, we estimated that the weights of the gelator-treated MECHS films were nearly half of the estimated dry weight (20.3% of wet pellet weight) of curli biomass (Supplementary Fig. 11c). Similarly, the weights of MECHS films obtained from 1% and 2% gelator were nearly 45% and 30%, respectively, of the estimated total weight of all precursors, whereas that for 3–5% gelator was about 25% (Supplementary Fig. 11d). These results also suggest that the convoluted MECHS films obtained from 1% and 2% gelator upon drying could be attributed to the incorporation of more cellular biomass into the films, while the 3–5% gelator might extract more cellular components like lipids into the supernatant (Supplementary Fig. 2b, c). Moreover, it is to be noted that unlike 1% and 2% of gelator concentrations, the 3–5% of gelator leads to better gelatinous curli biomass (Supplementary Fig. 1). As the percentage weight of MECHS with respect to the dry weight of curli biofilm

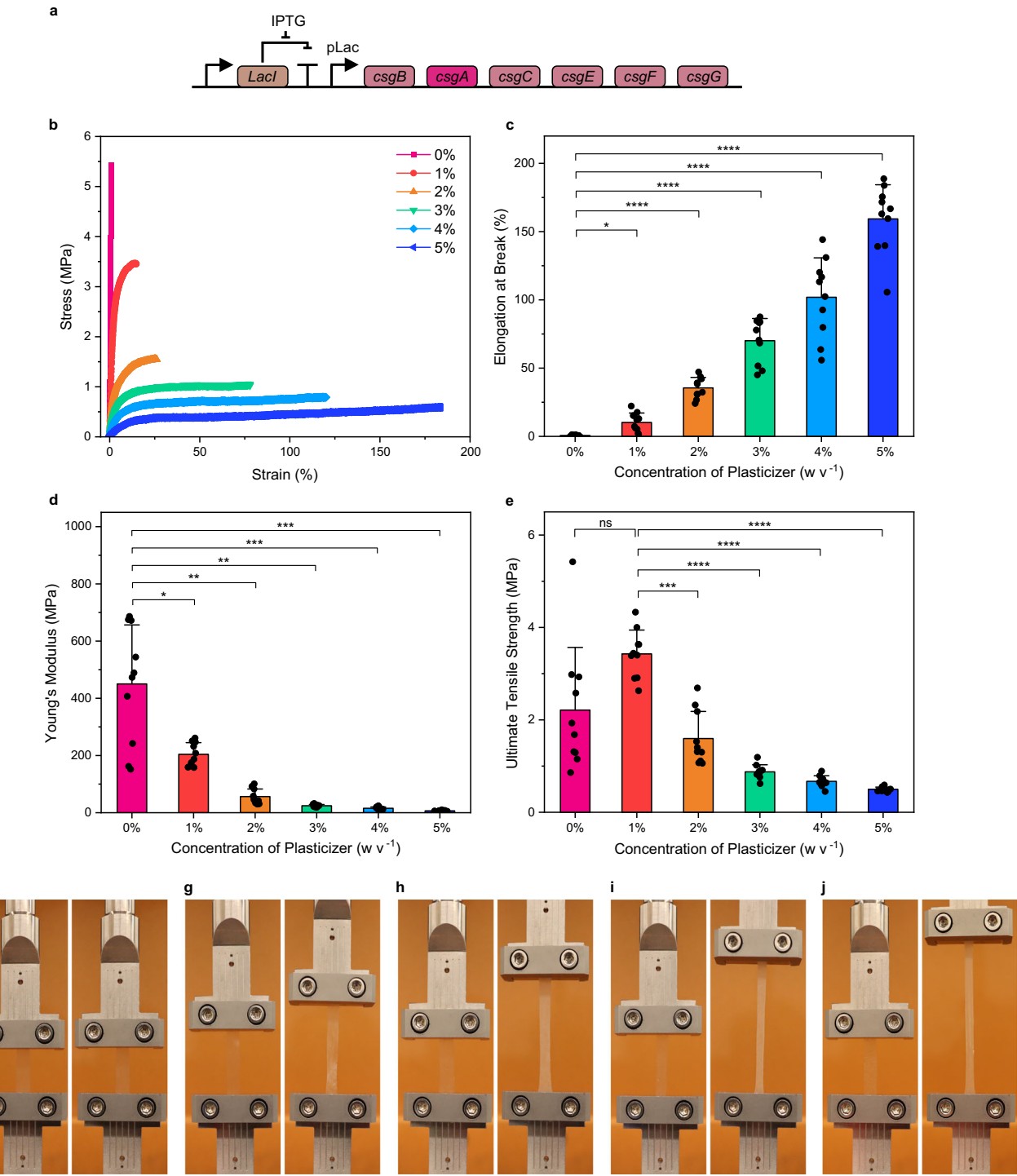

**Fig. 2 | Mechanical Properties of MECHS. a** Genetic design of *E. coli* to produce curli nanofibers. **b** Representative stress-strain curves of MECHS treated with 0 to 5% plasticizer. **c** Elongation at break, (**d**) Young's modulus, and (**e**) Ultimate tensile strength of MECHS treated with 0 to 5% plasticizer. Biological replicates $n = 10$. Data represented as mean ± standard deviation. **c** *$p = 0.0132$, ****$p < 0.0001$.

**d** *$p = 0.029$, **$p = 0.0021$, **$p = 0.0011$, ***$p = 0.0009$, ***$p = 0.0007$. **e** *$p = 0.1203$, ***$p = 0.0009$, ****$p < 0.0001$. One-way ANOVA followed by Tukey's multiple comparisons test. **f–j** Representative photographs of tensile tests of MECHS films with a lateral dimension of 0.5 cm by 4 cm. **f** 1%, (**g**) 2%, (**h**) 3%, (**i**) 4%, and (**j**) 5% of plasticizer. Left image: initial. Right image: before the break.

remains at around 45%, it suggests that the higher gelator (3–5%) content might not lead to additional loss of biomass into supernatant (Supplementary Fig. 11c). This latter inference is also supported by the fact that weight of dried supernatant increases in steps of ~50 mg, which is consistent with the expected increase in the theoretical weights of gelator (e.g., 5 ml of 1% accounts for 50 mg) (Supplementary Fig. 11b).

As noted above, 3–5% gelator-treated MECHS comprises nearly 45% dry weight of the whole cell pellet, then we reasoned that by determining the amount of SDS, we could estimate the total (cellular and curli) biomass in the MECHS (Supplementary Fig. 11c). By using Energy Dispersive X-ray Analysis (EDAX) we found out that for 3% gelator-treated MECHS, the weight percentage of Sodium and Sulfur elements were $2.2 \pm 0.2\%$ and $4.5 \pm 0.3\%$, respectively, whereas the

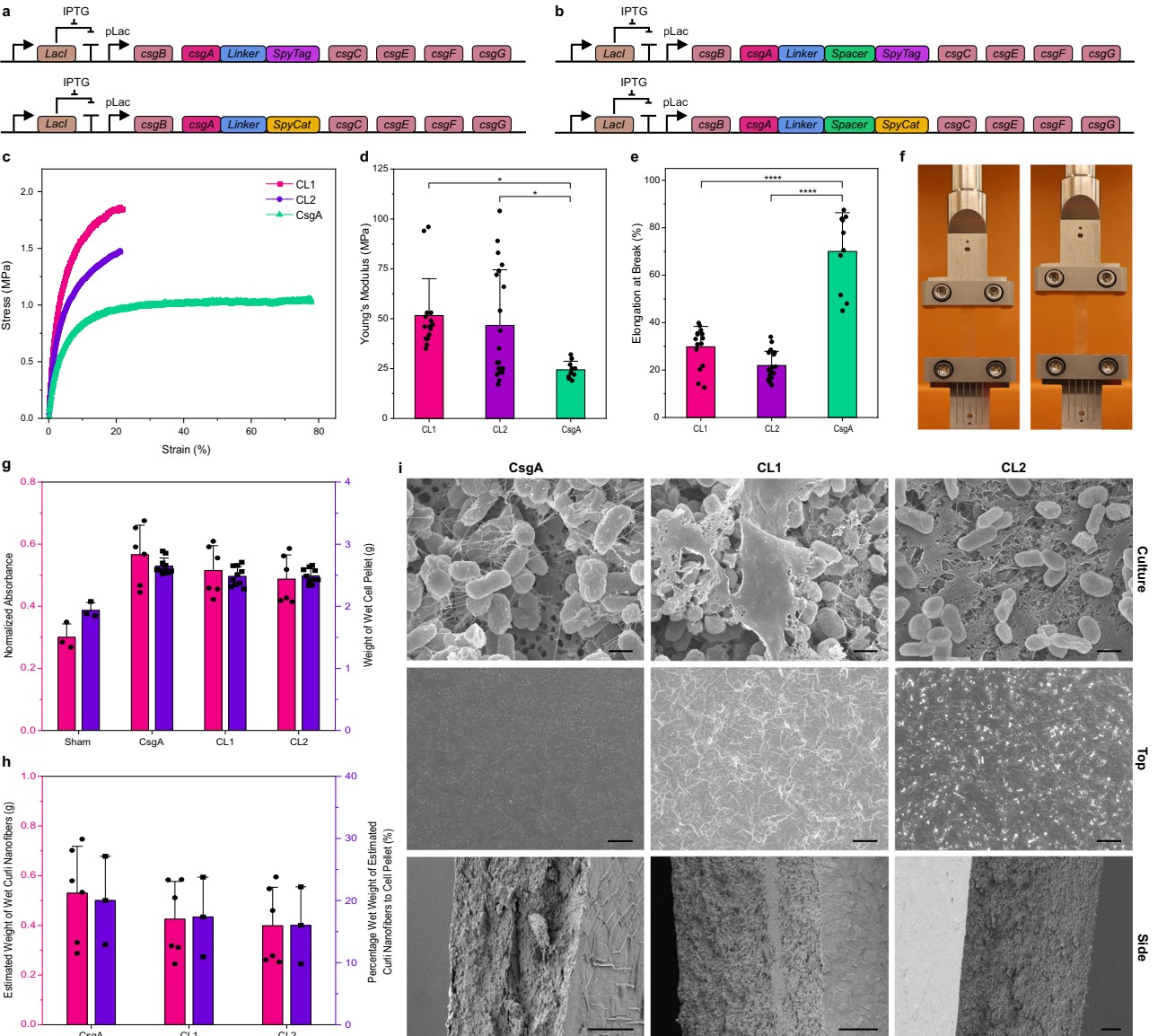

**Fig. 3 | Tailoring the Mechanical Properties of MECHS through genetic engineering.** Genetic design of *E. coli* to produce the functional curli nanofibers to covalently crosslink (**a**) CL1: SpyTag and SpyCatcher (SpyCat) domains fused to CsgA, (**b**) CL2: SpyTag and SpyCat domains fused to CsgA via the Spacer. **c** Representative stress-strain curves of MECHS films consisting of CsgA, CL1, and CL2 with 3% plasticizer. **d** Young's modulus, and (**e**) Elongation at the break for CsgA, CL1, and CL2 with 3% plasticizer. Biological replicates $n = 10$ for CsgA, $n = 15$ for CL1, and $n = 20$ for CL2. Data represented as mean ± standard deviation. **d** *$p = 0.01$, *$p = 0.0295$. **e** ****$p < 0.0001$. One-way ANOVA followed by Tukey's multiple comparisons test. **f** Representative photographs showing a tensile test of CL1 film with the lateral dimension of 0.5 cm by 4 cm. Left image: initial. Right image: before the break. **g** Plot of normalized Congo Red absorbance and the weights of wet cell pellets. For Congo Red absorbance, biological replicates $n = 3$ for Sham and $n = 6$ for CsgA, CL1 and CL2. For weights of wet cell pellets, biological replicates $n = 3$ for Sham and $n = 10$ for CsgA, CL1 and CL2. **h** Plot of the estimated wet weight of curli nanofibers and the wet weight percentage of estimated curli nanofibers to the cell pellet. Biological replicates $n = 6$ for the wet weight of curli nanofibers and $n = 3$ for the percentage weight. Data represented as mean ± standard deviation. **i** Field Emission Scanning Electron Microscopy (FESEM) images of CsgA, CL1 and CL2. Top row: cell cultures. Scale bar 1 µm. Middle row: Top view of MECHS. Scale bar 10 µm. Bottom row: Side view of MECHS. Scale bar 10 µm.

same elements for the curli biofilm cell pellet (without SDS treatment) were $0.6 ± 0.1\%$ and $1.2 ± 0.5\%$, respectively (Supplementary Fig. 13). Using this data, we estimate that for 3% gelator-treated films, roughly 5% (~ 1.6% Sodium and ~ 3.3% Sulfur) of MECHS weights could comprise of SDS (Supplementary Fig. 11a, c). Therefore, we can estimate that about 40% of the total cellular and curli biomass might be utilized to form the gelator-treated MECHS.

On the other hand, based on the weights of plasticizer-treated MECHS films and their corresponding dry supernatant weights, we could estimate that 15–20% of the total plasticizer utilized might get incorporated into MECHS, assuming that no additional biomass was lost to the supernatant during this phase of fabrication

(Supplementary Figs. 11, 14 and 15). In addition, the weights of MECHS films of CsgA, CL1, and CL2 and their dried supernatants were in the same range, which further validates that the covalent crosslinking in CL1 and CL2, leads to increased stiffness and not due to any variations in the plasticizer amounts (Supplementary Figs. 16 and 17).

Field Emission Scanning Electron Microscopy (FESEM) images from cultures of CL1, and CL2 showed aggregated mats of material, presumably due to nanofiber bundling promoted by the SpyTag/Spy-Catcher covalent crosslinking. Images obtained from CsgA cultures did not show such aggregation (Fig. 3i). FESEM images of MECHS (top and side view) further showed that the curli biomass is densely packed to form continuous films (Fig. 3i and Supplementary Figs. 18, 19).

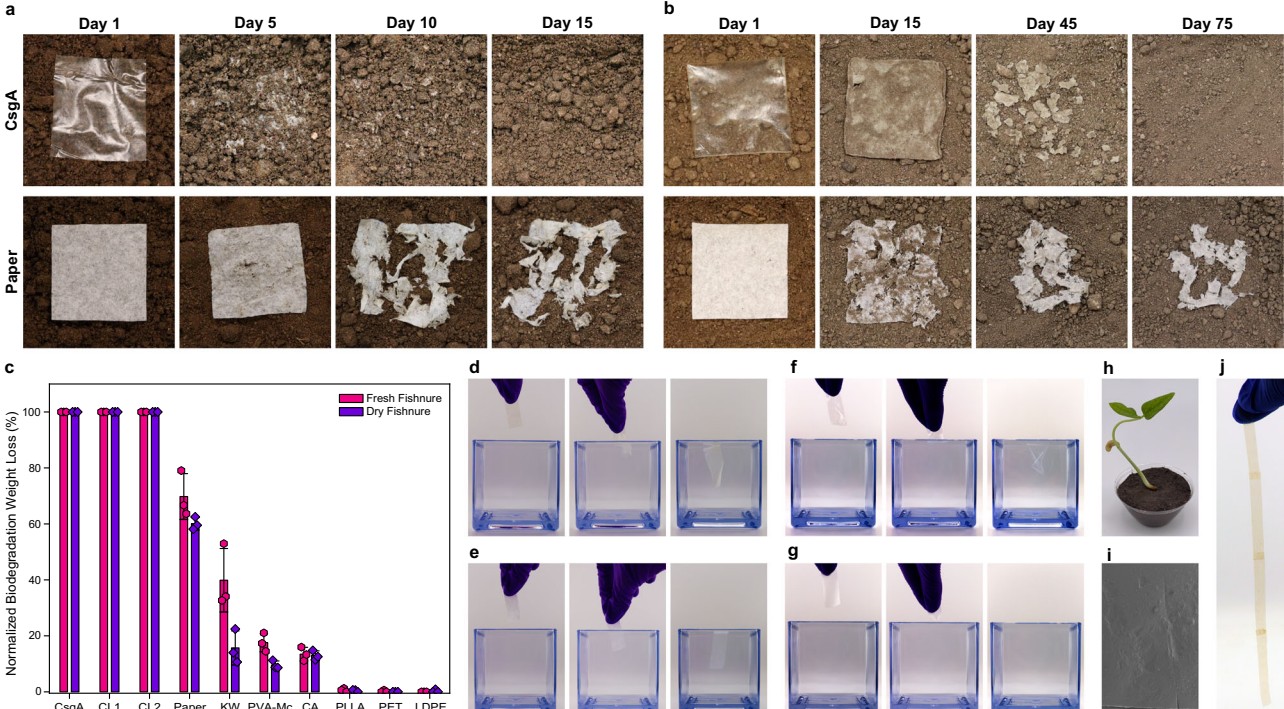

**Fig. 4 | Compostability, Water-dispersibility, and Healability of MECHS.** Representative photographs showing the biodegradation of MECHS and toilet paper in (**a**) a fresh fishnure (**b**) a dry fishnure. The lateral dimensions of the MECHS film and the toilet paper were 5 cm by 5 cm. **c** Plot shows the normalized biodegradation weight loss of MECHS (CsgA, CL1, and CL2), toilet paper, kimwipe (KW), polyvinyl alcohol - Mckesson (PVA-Mc), cellulose acetate (CA), poly-L-lactic acid (PLLA), polyethylene terephthalate (PET) and low-density polyethylene (LDPE). Biological replicates $n = 3$ for CsgA, CL1 and CL2. Technical replicates $n = 3$ for paper, KW, PVA-Mc, CA, PLLA, PET, and LDPE. Data represented as mean ± standard deviation. Photographs show the dissolution of **d** MECHS **e** toilet paper, **f** PVA-Mc, and **g** polyvinyl alcohol - Superpunch (PVA-Sp). **d**–**g** Lateral dimension of the films was 1 cm by 5 cm. **h** Photograph of a black bean seedling grown in the soil mixed with fishnure (comprising biodegraded MECHS) in a 9:1 ratio. **i** FESEM image of MECHS film healed by placing microliters of water at the site of abrasion (black arrows). Scale bar 200 μm. **j** Photograph shows the MECHS films welded (black arrows) by using water. Scale bar 0.5 cm.

## Compostability, scalability, and mechanical landscape

To test the relative compostability of MECHS films compared to other conventional plastics and bioplastics, we buried samples of each in a commercially available compost called fishnure, derived from fish manure. Experiments were performed in a mini greenhouse setup with samples of uniform size and shape (Supplementary Figs. 20, 21). Under these conditions, MECHS films biodegraded completely in 15 days, while all the other samples did not (Fig. 4a, c and Supplementary Figs. 21–23). Toilet paper and kimwipes biodegraded to 70% and 40%, respectively (Fig. 4a, c and Supplementary Fig. 21). The bioplastics, cellulose acetate (CA), and poly-L-lactic acid (PLLA) were biodegraded by 13% and 1% respectively, whereas the petrochemical plastics polyethylene terephthalate (PET) and low-density polyethylene (LDPE) did not show any biodegradation (Supplementary Fig. 22). On the other hand, two different commercial polyvinyl alcohol (PVA) formulations, PVA-Mc and PVA-Sp, lost 17% weight and completely disappeared in 5 days, respectively (Supplementary Fig. 23).

Some of the mass loss in the experiments above may have been attributable to dissolution in the moist fresh fishnure, rather than biodegradation, especially for MECHS and PVA. Therefore, we performed additional compostability tests in fishnure that was dried (i.e., placed in the greenhouse for 50 days). Under these conditions, MECHS films were able to biodegrade completely in 75 days (Fig. 4b, c and Supplementary Fig. 24). The toilet paper, kimwipe, and CA were found to degrade by about 60, 16, and 13%, respectively, whereas PLLA, PET, and LDPE did not show any biodegradation in dry fishnure (Fig. 4b, c and Supplementary Figs. 24, 25). However, PVA-Mc had nearly 10% weight loss, whereas PVA-Sp was found to be intact even after 75 days in dry fishnure. We could not determine the weight loss of PVA-Sp as the film was firmly sticking to the fishnure granules. These experiments show that the MECHS films are completely compostable and that their biodegradation compares favorably to many plastics, bioplastics, and even toilet paper.

For the potential use of MECHS as flushable packaging materials, we tested its ability to dissolve in water (Fig. 4d, e). The MECHS films did not dissolve completely, likely due to the network of hydrophobic curli nanofibers. We speculate that the more water-soluble components like glycerol, SDS, and the other cellular biomass leach into water more readily. PVA-Sp dissolves completely in water, whereas PVA-Mc dissolves only partially, leaving behind water-insoluble strips possibly compromising its biodegradation (Fig. 4f, g). Furthermore, except for MECHS, all the other plastics compared here are composed only of carbon, hydrogen, and oxygen. Therefore, their biodegradation is often considered, in terms of breakdown completely, to lead to carbon dioxide. However, MECHS is largely composed of protein, making it the only plastic amongst those compared here with any significant nitrogen content. Therefore, it may be reasonable to consider its potential as a biofertilizer to support plant growth (Fig. 4h). Further, MECHS films could also be healed and welded by using microliters of water at the site of abrasion or attachment and subsequently ambient dried (Fig. 4i, j).

The fabrication method presented in this work yielded 500–1000 mg of MECHS films per liter of culture, which is nearly 10 times higher than the 50–100 mg obtained from our previously reported AquaPlastic protocol[9]. In addition, the MECHS biofabrication method speeds up the process in comparison to the tedious and slow filtration process utilized for AquaPlastic. We achieved these yields even with a standard shake-flask format that is routinely used in laboratory settings for recombinant protein production. Therefore, tens of liters of bacterial culture could be used to fabricate large MECHS prototypes, such as thin films, tens of centimeters in one

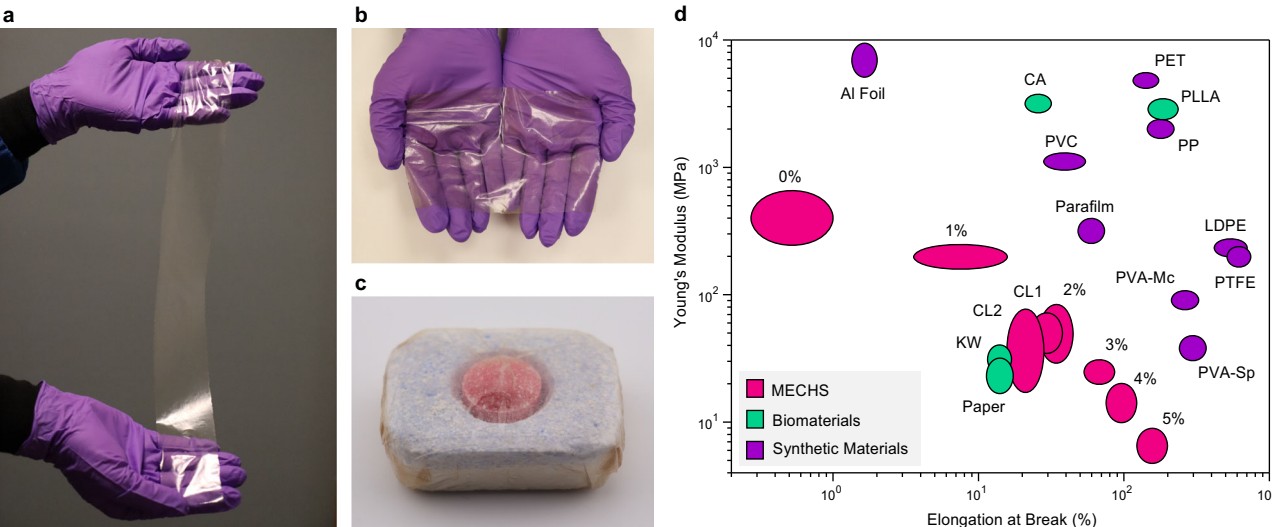

**Fig. 5 | Prototypes and Mechanical Landscape of MECHS. a** Photograph shows a refined prototype of MECHS film with a lateral dimension of 5 cm by 50 cm. **b** Photograph shows the optical transparency of MECHS film with a lateral dimension of 10 cm by 15 cm. **c** Photograph of a detergent pod (lateral dimension of 4 cm by 3 cm) wrapped with a MECHS film. **d** Ashby plot shows Young's modulus and elongation at break for MECHS and various synthetic materials and biomaterials. Low-density polyethylene (LDPE), Polytetrafluoroethylene (PTFE), Poly-L-lactic acid (PLLA), Polyethylene terephthalate (PET), Cellulose acetate (CA), Polypropylene (PP), Polyvinyl chloride (PVC), Polyvinyl alcohol - Superpunch (PVA-Sp), Polyvinyl alcohol - Mckesson (PVA-Mc), Aluminum foil (Al Foil), Parafilm, Kimwipes (KW) and Toilet paper.

dimension (Fig. 5a, b and Supplementary Fig. 26). We also created a detergent pod as an example of the flushable and biodegradable primary package (Fig. 5c).

To better visualize and compare the mechanical properties of MECHS, we present an Ashby plot of Young's modulus and elongation at break for various plastics, bioplastics, biomaterials, and synthetic materials (Fig. 5d and Supplementary Figs. 27–35). It is thus evident that Young's modulus of MECHS is in the same range of LDPE, PTFE (polytetrafluoroethylene), PVA, and paper, while its elongation at break matches that of CA, PVC (polyvinyl chloride), PP (polypropylene), PET, PLLA, and parafilm.

## Discussion

We previously reported the curli nanofiber-based bioplastic fabrication protocol (i.e., AquaPlastic), which involved the filtration of bacterial culture to concentrate curli nanofibers and form gels[9]. Using that protocol, concerns about clogging necessitated the use of filters with 10 μm pores, leading to the loss of significant amounts of curli nanofibers. The MECHS fabrication protocol described in this paper increased the yield of bioplastic by a factor of ten by utilizing not only all the curli nanofibers in the pelletized biomass but also the other water-insoluble cellular biomass. We also found that the SDS gelator could be supplemented with a plasticizer like glycerol to obtain flexible films of MECHS, as compared to the significantly more brittle Aqua-Plastic. Glycerol, being a byproduct of the biodiesel industry, offers several advantages *viz.*, nontoxic, low-cost, and renewable[45]. Unlike the conventional petrochemical plastics and other bioplastics that are processed by thermal molding, MECHS was molded into films by ambient drying of gelatinous biomass, which we have termed aqua molding. The healing and welding of MECHS films by using tiny droplets of water are termed aqua healing and aqua wedding, respectively.

The tunability of MECHS, with its range of mechanical properties (e.g., elongation at break 1–160%; Young's modulus 6–450 MPa) and transparency, provides a promising platform to access biodegradable alternatives to synthetic materials like petrochemical plastics. We were also able to use our streamlined protocol to achieve high yields of 0.5–1 g L$^{-1}$ and generate large, refined prototypes. Further, we could obtain ~ 40 cm$^2$ of MECHS film per liter of culture. Therefore, to biofabricate a roll of MECHS film with the lateral dimensions 2 m × 5 cm, we would require ~ 25 L of culture. Another notable feature of this work is that 40% of the total cellular biomass gets incorporated into the plastic/paper-like MECHS, which could also be instrumental in attracting further research to utilize cellular biomass for the development of various sustainable functional materials.

During the MECHS biofabrication, most of the SDS ends up in the supernatant, which, when dried, resulted in a brown-yellowish color pellet. Thus, we believe that SDS, being a surfactant removes the brown-yellowish color of the cell pellet, which makes MECHS film transparent. Curli nanofibers are assembled from CsgA protein building blocks that comprise a rigid beta-helical structure, which, in simple terms, can be regarded as a quasi-crystalline ordering. So, when these curli nanofibers (without plasticizer) based rigid materials are subjected to tensile stress beyond its yield point, the strain-induced crack propagates, and it quickly breaks the material. However, by incorporating a plasticizer like glycerol, the amorphous nature of the plasticizer that surrounds the rigid curli nanofibers inhibits crack propagation by subjecting the material to undergo plastic deformation. Thus, with increasing plasticizer amounts from 1–5%, the elongation at break was found to increase from 1 to 160%.

Plastics are one of the most abundant human-made materials, with over 8.3 billion tons produced cumulatively, 79% of which are estimated to have accumulated in landfills and oceans[46]. In addition, the contamination of microplastics in almost all parts of the globe further enhances their threat to our health and the environment[47,48]. Biodegradable bioplastics account for less than 1% of the global plastic market, and their limited properties warrant the development of alternatives[35]. Given that the typical lifetime of packaging material is 1-2 years, and the packaging industry accounts for nearly one-third of the plastic market, there exists a tremendous scope and opportunity for biodegradable packaging, though success will likely need to be achieved through the commercialization of drop-in replacements for existing materials. Notably, water-soluble polymers like PVA (commonly found in detergent pods) have limited biodegradation under diverse settings of land and water[49]. In many cases, dissolvable

polymers like PVA are blended with petrochemical plastics to enhance certain material properties, but this limits their water dispersibility and biodegradability (as observed in our biodegradation tests with the commercially available PVA-Mc)[50].

Although we were able to develop refined prototypes of MECHS thin films, additional work will be needed to improve the mechanical properties (e.g., ultimate tensile strength, tear strength) and resistance to water. Furthermore, the circular materials economy loop will have to be closed by employing a feedstock for bacterial culture derived closely from $CO_2$ fixation, such as cellulose hydrolysate obtained from agricultural waste. There are also several opportunities to utilize synthetic biology tools to tailor the material properties of curli nanofibers, which need to be explored. The concept of using biodegraded MECHS as a biofertilizer for plant growth warrants further investigation. All in all, in this work, we have demonstrated that the manufacturing capabilities of living cells can be employed to produce mechanically tunable, scalable, and compostable ELMs as a potential alternative to synthetic materials like plastics. Finally, we believe that innovative approaches involving synthetic biology and materials engineering could lead to greater advancements in creating energy-efficient and sustainable solutions to a greener ecosystem.

## Methods

### Materials
Low-density polyethylene LDPE (ET31-FM-000151, 50 μm thick), Polytetrafluoroethylene PTFE (FP30-FM-000250, 50 μm thick), Poly-L-lactic acid PLLA (ME33-FM-000150, 50 μm thick), Polyethylene terephthalate PET (ES30-FM-000150, 50 μm thick), Cellulose acetate CA (AC31-FM-000151, 50 μm thick) and Polypropylene PP (PP30-FM-000250, 50 μm thick) were obtained from Goodfellow Corporation. Polyvinyl chloride PVC (S-16280, 15 μm thick) was obtained from Uline. Polyvinyl alcohol - Superpunch PVA - Sp (ASIN: B01M11T6U5; a water-soluble stabilizer for embroidery and it is claimed to be made from 100% PVA and thus it dissolves completely in water), Polyvinyl alcohol - Mckesson PVA - Mc (ASIN: B01ETFMUH2; a hot water soluble bag, which is also claimed to be made from PVA, but it does not dissolve completely in water at room temperature, probably because PVA might be blended with other components.) and Silicone mats (ASIN: B09SPB72TT) were obtained from Amazon. Aluminum foil, Parafilm, Toilet paper, and Kimwipes were obtained from Reynolds Consumer Products, Bemis Company Inc., Signature Select, and Kimberly-Clark Corporation, respectively. Glycerol (G9012) and Sodium dodecyl sulfate SDS (S0295) were obtained from Sigma-Aldrich and Teknova, respectively.

### Plasmids to produce MECHS.
pET21d plasmid was cloned with the curli operon genes *csgA*, *csgB*, *csgC*, *csgE*, *csgF*, and *csgG* that encodes the proteins necessary for the biosynthesis of curli nanofibers and it is labeled as pET21d-CsgA. The genes encoding the SpyTag peptide and SpyCatcher protein derived from an earlier report[43] were fused to the C-terminus of CsgA with an intervening 36 amino acid flexible linker to obtain the plasmids pET21d-CsgA-SpyTag and pET21d-CsgA-Spy-Catcher, respectively (Supplementary Table 1). The gene encoding the Spacer, an intrinsically disordered protein[44], was inserted between the linker and the SpyTag or the SpyCatcher to obtain pET21d-CsgA-Spacer-SpyTag and pET21d-CsgA-Spacer-SpyCatcher, respectively. The genes were synthesized (Integrated DNA Technologies) and cloned into pET21d vector using isothermal Gibson assembly (New England Biolabs).

### Cell strain to produce MECHS.
The plasmids pET21d-CsgA, pET21d-CsgA-SpyTag, pET21d-CsgA-SpyCatcher, pET21d-CsgA-Spacer-SpyTag, and pET21d-CsgA-Spacer-SpyCatcher were separately transformed into PQN4, an *E. coli* cell strain derived from LSR10 (MC4100, Δ*csgA*, λ(DE3), Cam^R) with the deletion of the curli operon (Δ*csgBACEFG*) to produce the corresponding MECHS[41].

### Cell culture to produce MECHS (CsgA).
pET21d-CsgA plasmid was transformed into PQN4 and streaked onto a lysogeny broth (LB) agar plate containing 100 μg ml⁻¹ carbenicillin and 0.5% glucose (m v⁻¹) for catabolite repression of T7RNAP and incubated overnight at 37 °C. A single colony of PQN4-pET21d-CsgA was picked from the agar plate and cultured at 37 °C in 5 ml LB media, 100 μg ml⁻¹ carbenicillin, and 2% glucose (m v⁻¹). The overnight culture was transferred to a fresh 500 ml LB media containing 100 μg ml⁻¹ carbenicillin and cultured for 48 h in incubator shakers (225 rpm, 37 °C) to express the CsgA curli protein nanofibers.

### Cell culture to produce the covalently crosslinked MECHS (CL1 and CL2).
For Covalently Crosslinked-1 (CL1), the plasmids pET21d-CsgA-SpyTag and pET21d-CsgA-SpyCatcher were separately transformed into PQN4 and streaked onto lysogeny broth (LB) agar plates containing 100 μg ml⁻¹ carbenicillin and 0.5% glucose (m v⁻¹) for catabolite repression of T7RNAP and incubated overnight at 37 °C. A single colony was picked from the agar plates of PQN4-pET21d-CsgA-SpyTag and PQN4-pET21d-CsgA-SpyCatcher and cultured separately at 37 °C in 5 ml LB media, 100 μg ml⁻¹ carbenicillin and 2% glucose (m v⁻¹). The overnight cultures of PQN4-pET21d-CsgA-SpyTag and PQN4-pET21d-CsgA-SpyCatcher were transferred to a fresh 500 ml LB media containing 100 μg ml⁻¹ carbenicillin and co-cultured for 48 h in incubator shakers (225 rpm, 37 °C) to express and covalently crosslink the engineered curli protein nanofibers. Similarly, the plasmids pET21d-CsgA-Spacer-SpyTag and pET21d-CsgA-Spacer-SpyCatcher were utilized for Covalently Crosslinked-2 (CL2) MECHS.

### Biofabrication of MECHS.
The 48 h cell culture (500 ml) of PQN4-pET21d-CsgA (CsgA) was centrifuged (5000 × *g*, 10 min) to pelletize the curli biomass, which was then washed with 250 ml of deionized water by centrifuging (5000 × *g*, 10 min) to remove the residual quantities of culture media. 1 × *g* (wet pellet) of curli biofilm biomass was first dispersed in 5 ml of deionized water and subsequently added with 5 ml of 1, 2, 3, 4, or 5% (w v⁻¹) of sodium dodecyl sulfate (SDS, serves as a gelator and also helps to obtain the transparent MECHS films by taking away the brown-yellowish color of cell pellet into the supernatant), which was then mixed on a shaker for 2 h at room temperature. The resulting gelatinous biomass was washed with 10 ml of deionized water twice by centrifuging (5000 × *g*, 10 min) to remove the soluble biomolecules and the excess SDS. This SDS treated gelatinous biomass was casted and ambient dried on a silicone mold to obtain the MECHS films that were brittle.

To realize the flexible films of MECHS, the 3% SDS-treated gelatinous biomass of PQN4-pET21d-CsgA (CsgA) was added with 5 ml of 1, 2, 3, 4, or 5% (w v⁻¹) of glycerol (serves as a plasticizer) and mixed on a shaker for 1 h at room temperature. The glycerol treated and centrifuged (5000 × *g*, 10 min) biomass was casted on a silicone mold and ambient dried to obtain the flexible MECHS films.

Similarly, to realize the Covalently Crosslinked (CL1 and CL2) films of MECHS, 5 ml of 3% SDS and 5 ml of 3% glycerol treated curli biomass was utilized. For all constructs, a minimum of ten replicates were tested.

### Field-emission scanning electron microscopy (FESEM) sample preparation and imaging.
100 μL of cell culture was vacuum filtered on a membrane (0.22 μm pore size, Millipore GTTP02500) and washed with 100 μL of deionized water thrice. The samples were fixed by immersing in 2 ml 1:1 mixture of 4% (w v⁻¹) glutaraldehyde and 4% (w v⁻¹) paraformaldehyde at room temperature, overnight. The samples were gently washed with water, and the solvent was gradually exchanged to ethanol (200 proof) with an increasing ethanol 15-minute incubation step

gradient [25, 50, 75, and 100% (v v$^{-1}$) ethanol]. The samples were then dried in a critical point dryer, placed onto SEM sample holders using silver adhesive (Electron Microscopy Sciences) and sputtered until they were coated in a 10–20 nm layer of Pt/Pd. Whereas the films of MECHS were directly sputter coated with a 10–20 nm layer of Pt/Pd without critical point drying. Images were acquired using a Zeiss Gemini 360 FESEM equipped with a field emission gun operating at 5–10 kV. Representative images from three independent samples were reported.

**Energy dispersive x-ray xnalysis (EDAX).** Oxford Instruments Ultim Max EDS equipped with AZtecLive software attached to Zeiss Gemini 360 FESEM was utilized to detect the elements as well as determine their composition using factory standards. EDS spectra were recorded on sample's surface with the lateral dimensions of 225 μm by 170 μm. Data from three independent samples were reported.

**Optical images.** Optical images were acquired using a Canon EOS Rebel SL3 Digital SLR Camera equipped with XIT 58 mm 0.43 Wide Angle Lens and XIT 58 mm 2.2x Telephoto Lens. Representative images from three independent samples were reported.

**Tensile tests.** Tensile measurements of MECHS, commercially available plastics, bioplastics, and all other materials mentioned in this report were performed using a DHR-3 rheometer (TA Instruments) under ambient laboratory conditions. Films with lateral dimensions of 4 cm by 0.5 cm under a constant linear deformation of 1 μm s$^{-1}$ were utilized for tensile tests. A minimum of five samples were tested for each type.

**Film thickness.** The thickness of the films was measured using a contact profilometer, Dektak 3ST equipped with a 2.5 μm stylus having a vertical resolution of 1 Å. A minimum of three tests were performed for each sample.

**Large prototypes.** The MECHS prototype of 50 cm × 5 cm lateral dimension was fabricated from 6 L cultures of PQN4-pET21d-CsgA (obtained by using 3% SDS and 3% glycerol treatment), whereas the 15 cm × 10 cm and the detergent pod prototypes were obtained from that of 4 and 3 L cultures, respectively.

**Healing.** The films of MECHS (PQN4-pET21d-CsgA, obtained by using 3% SDS and 3% glycerol) were cut using scissors, and ~10 μL of deionized water was added at the cut site and subsequently dried at ambient laboratory conditions to heal the cut. A minimum of three samples were tested. Similarly, MECHS films of 0.5 cm by 5 cm were welded by using ~10 μL of deionized water and subsequently dried at ambient laboratory conditions.

**Biodegradation.** A commercially available odorless organic humus compost named Fishnure (Amazon, ASIN: B086KXT5TQ), which is made from fish manure, was utilized for the biodegradation test. Samples with lateral dimensions of 5 cm × 5 cm were buried in a tray containing 3.5 kg of Fishnure. The biodegradation experiment was conducted in a mini greenhouse (Amazon, ASIN: B01D7GHEES) setup (exposed to direct/indirect sunlight through the large windows of the laboratory), wherein a temperature of 20 °C and a relative humidity of 80% was maintained. The films of MECHS degraded completely in 15 days in a freshly opened bag of Fishnure. In another biodegradation experiment, a dry (by placing in the mini greenhouse setup for 50 days) Fishnure was utilized and under these conditions, films of MECHS degraded completely in 75 days. A minimum of three samples were tested for each type.

**Congo Red assay.** 1 ml of cell culture (as described above: 48 h, 500 ml at 37 °C) was pelleted by centrifuging (6000 × g, 10 min), and the resulting cell pellet was incubated with 1 ml of 0.004% (w v$^{-1}$) Congo Red dye for 10 min. The dye-treated cell culture was pelletized by centrifuging (6000 × g, 10 min), and the resulting supernatant (200 μL) was utilized to measure the absorbance at 480 nm in a plate reader. The net Congo Red absorbance of curli in CsgA, CL1 and CL2 were determined by subtracting the absorbance values of cell pellet having a sham plasmid (without curli operon), to account for the non-specific binding to other biomolecules.

To estimate the curli nanofibers produced, we utilized 0.004% (w v$^{-1}$) Congo Red dye to prepare a standard curve for various concentrations of purified CsgA. Herein, C-terminal His-tagged CsgA (CsgA-His) was expressed and purified using Ni-NTA (Nickel-nitrilo-triacetic acid resin) column[51]. However, after eluting the CsgA-His with the elution buffer, the buffer was exchanged with water using a 10 kDa Amicon centrifugal filter. This buffer exchange facilitates fibrillation of CsgA-His and the resulting pellet (wet weight) was utilized for the CsgA (CsgA-His) Congo Red standard curve. A minimum of three samples were tested for each type.

**Statistics and reproducibility.** All experiments presented in this article were repeated at least three times ($n \geq 3$) on distinct samples or biological replicates, as clearly specified in the figure legends or the relevant Methods sections. In all cases, data are presented as the mean and standard deviation. GraphPad PRISM 8, OriginPro 2024, Oxford Instruments Ultim Max EDS AZtecLive software, TRIOS software V5.2, Adobe Photoshop 2024 and Adobe Illustrator 2024 were utilized for plotting and analyzing data. For micrographs and optical images, we present representative images.

### Reporting summary
Further information on research design is available in the Nature Portfolio Reporting Summary linked to this article.

## Data availability
All relevant data supporting the findings of this study are available within the Article and its Supplementary Information. Source data are provided in this paper.

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

## Acknowledgements

Work was performed in part at the Center for Nanoscale Systems at Harvard University and George J. Kostas Nanoscale Technology and Manufacturing Research Center at Northeastern University. Work in the N.S.J. laboratory is supported by the National Science Foundation, USA (DMR 2004875) and Novo Nordisk Foundation Challenge Program 2022 - Energy Materials with Biological Applications (NNF22OC0071130). We thank Arjun Rajesh and Bismay Hota for their assistance with capturing photographs of MECHS biodegradation and prototypes. Parts of the schematics were adapted from BioRender.com.

## Author contributions

A.M.-B. conceived the project and performed all the experiments and analyses. A.M.-B. and A.M.D.-T. cloned all the curli variants. A.M.-B. and N.S.J. wrote and edited the manuscript. All authors discussed and commented on the manuscript.

## Competing interests

A.M.-B., A.M.D.-T., and N.S.J. are inventors on a U.S. Provisional Patent Application (63/604,497) submitted by Northeastern University. All authors declare no other competing interests.
