## [Peer Review File · Nature Communications]

Mechanically Tunable, Compostable, Healable and Scalable
Engineered Living MaterialsREVIEWER COMMENTS

Reviewer #1 (Remarks to the Author):

Manjula-Basavanna, Joshi, and colleagues reported a new process for engineered living materials (ELM) from cellular biomass and engineered curli fibers. Building on their previous work in Aqua plastics, they utilized 1–5% glycerol as a plasticizer to obtain more ductile ELMs in higher yields. The main innovation in this paper is the process of converting cellular biomass and engineered proteins into robust, tangible, and scalable materials comparable to plastics and papers, overcoming the well-known problem of brittleness of dried ELM. I have a few questions and comments for this paper, as noted below. I recommend considering acceptance of this article after the authors could respond to these points fully.

Congo Red (CR) staining provides a qualitative result and may generate a few different artifacts. According to the SI, the authors treated cell culture with CR solution, removed cells by centrifugation, and measured the absorbance of supernatant at 480 nm. The data was compared with the standard curve prepared with purified CsgA (Figure S9). First of all, I do not understand the rationale for using absorbance values at 410 nm instead of fluorescence, which is more characteristic of the CR in a restricted environment such as amyloid. A previous paper from the Joshi lab (Applied and Environmental Microbiology 2019) describes a higher absorbance reading of CR staining in cell-only samples compared to cell and curli fiber samples (Figures 1 and 2 in the AEM paper), highlighting why fluorescence is more useful for quantifying the amyloid formation. The 410 nm reading will likely give signals from all of the CR molecules in the supernatant, including free-floating CR and non-specifically absorbed CR to the curli fibers (or other biomacromolecules). Moreover, while the standard is prepared with a pure CsgA protein, the staining of CR was done in a more complex mixture, raising questions about whether this is an accurate estimation of the amount of curli fibril. In addition, the standard curve in Figure S9 yielded a linear response between 0.2–0.4 absorbance (corresponding to 0.8–1.2 mg/mL purified protein), but the data from the sample is completely out of that linear range (absorbance around 1.0). We also do not know whether a known number of cells mixed with a known amount of curli fiber would give accurate results with this method. As an alternative to the CR staining, the authors may be able to use immunoblots to estimate the amount of curli fiber in their system.

The authors also mentioned that the amount of curli fiber species from 500 mL cultures is estimated to be 400–500 mg. Since their estimation method compares the CR staining to the standard solution prepared with a known dry weight of protein (because there is no way to measure the wet weight of protein...), these numbers are the dry weights of individual curli fibers. Also, the wet weight of the pellet obtained in 500 mL culture is ~2500 mg. For *E. coli*, the wet weight is around 1.7 g/L, whereas the dry weight is around 0.39 g/L (23% of the wet weight according to Bionumbers, somewhat less than this but also supported by their dry weight estimation in Figure S10). If their claim is true, their cell culture contains 500 mg of curli protein in 575 mg of cellular biomass (23%

of the wet weight, likely less considering their Figure S10), leaving only ~75 mg of everything else in cells. They mentioned that the wet weight of the control (sham plasmid) is ~1900 mg, which corresponds to approximately 437 mg of cellular biomass in dry weight. It does not add up. This is one of the reasons why I think their estimation of curli fiber is significantly off.

They quantified the amount of SDS in the gelatinous biomass by EDAX and compared the dry weights of the gel and supernatant. It looks like a significant portion of the biomass is being lost to the supernatant during this process, leaving only 45% of dry weight (whatever is left plus SDS). They took the amount of sodium and sulfur to calculate the amount of SDS in the gel. I don't think this method makes a lot of sense when a significant portion of biomolecules/cellular mass is being lost to the supernatant during this step. Sodium and sulfur elements also exist in the original biomass and cannot give characteristic contrast to enable accurate estimation of SDS.

It appears that the amounts of curli proteins expressed (by weight) are similar among CsgA, CL1, and CL2. Curli proteins in CL1 and CL2 are 40–120% and 200–280% longer (and heavier) than CsgA proteins. Assuming the similar expression level of these proteins, the effect of SpyTag-SpyCatcher crosslinking will be negated by the less amount of CsgA units and decreased amount of assembly. If the weights of MECHS films of CsgA, CL1, and CL2 were in the same range (as shown in Figure S16), why do you see the increase in Young's modulus at all? Are there any differences in expression levels of the curli fiber in these systems?

How many cells in a given weight or volume are present after SDS and glycerol treatment? I am also curious to know why this material is transparent when (some of the) cells are still intact within the materials. A more detailed investigation of gels with SDS and glycerol (rheology, SEM) before drying can provide more insights into compositions and crosslinking.

Why does adding plasticizers change the mechanical properties of materials? More molecular explanation and systematic investigation of other plasticizing compounds will provide clues as to why the resulting MECH properties.

It is unclear how PVA-Mc and PVA-Sp are different.

Reviewer #2 (Remarks to the Author):

In this study, the authors reported Mechanically Engineered Living Material with Compostability, Healability, and Scalability (MECHS) in the form of a stretchable plastic, with the feature of biodegradability, manufacturability, sustainability, and ability to tailor functional properties. The MECHS is produced directly from cultured bacterial biomass, and the fabrication protocol described in this paper increased the yield of bioplastic significantly. The highlight of the work relies on the mechanical properties and biodegradability. Overall, I feel that it could merit publication

after minor revision and some necessary revision.

Major issues:

- (1) The authors only mentioned “tried analogous experiments with a large spacer (disordered protein domain of 225 amino acids) in between CsgA and the SpyTag/SpyCatcher domains” on Page 4. Further discussion to explain the reason for inserting IDP between CsgA and the SpyTag/SpyCatcher domains in more detail is encouraged.
- (2) The authors mentioned, “Bigger spacer domain might lead to significant reductions to stiffness and enhanced extensibility.” Have the authors tried proteins with large well-folded structures as Spacers, and what are the expected properties?
- (3) The authors mentioned the process of biofabrication of MECHS in SI where the 48 h cell culture was centrifuged, washed, and subsequently added SDS, etc. Further discussion about the comparisons of this approach to other manufacturing methods would be encouraged (ex. the operation difficulty, time, and yield).
- (4) The authors demonstrated the MECHS films were able to biodegrade completely in 75 days in fishure, showing completely compostable features. I am curious about the durability of this material in its use scenarios, would it degrade on its own in use scenarios? Please comment on this.

Minor issues:

- (1) The left part of Figure 1 is not clear enough. It seems like three different kinds of bacteria are co-cultured in the same culture flask. The authors are suggested to modify it to avoid misunderstanding. SpyTag's sequence is much shorter than SpyCatcher's, and the legend of the unfolded peptide should be modified to the actual length.
- (2) Please make sure that the upper and lower case in the title of Materials and Methods is uniform.

Reviewer #3 (Remarks to the Author):

The manuscript by Duraj-Thatte and Manjula-Basavanna describes a new E.coli-based ELM called MECHS that consists of dried E.coli cell mass, plasticized with small amounts of glycerol and formed into transparent flexible films that biodegrade readily in compost. Through genetic engineering of the E.coli to produce curli fibres in abundance, the team improve and tune key mechanical properties of their materials. They can also tune properties by changing the glycerol content and use of SDS in turning the cell pellet into a material that can become a film. Finally, the coolest bit of work of all, is having 2 E.coli strains mixed together that produce engineered curli fibres that either have spytag or spycatcher proteins all along them. This means when the different bacteria meet in the biomass their fibres click together, increasing the material strength. The paper has lots of nice materials testing, some very convincing biodegradation experiments and even makes a film to wrap a dishwasher tablet. Cool stuff.

Overall, I really liked this paper and thought it was very well written and presented. I'm not sure if it's been reviewed at another journal beforehand, but it felt like this was a version that had been read, edited and double-checked several times before. I struggled to find anything to correct.

In principle I'm happy for this to be published as it is - it's very good work and good quality.

My only three comments would be:

1. What is the effect of the culture media composition on the material? i.e. if the E.coli grow in LB in a shake flask, do they make a different quality/strength of material than if they grow on a mixed waste source in a fermentor, instead?
2. What is the amount of E.coli liquid culture needed to grow to make 10 cm sq and 1 metre sq? Is it feasible to make a roll of film (e.g 2m long, 5 cm wide) similar to sellotape just by growing a few flasks of bacteria? It would be good to get an idea of scale of production needed to make some amounts that would match real-world use cases.
3. The entire two page section 'composition and morphological analysis' was difficult to get through and rather boring. Could the authors consider shortening it or revising it to get to the key points more quickly? I felt like it took forever to get to any key messages from this part of the study.

RESPONSE TO REVIEWER COMMENTS

Nature Communications manuscript NCOMMS-24-08057-T “Mechanically Tunable, Compostable, Healable and Scalable Engineered Living Materials”

* Original reviewer comments in regular typeface; **author responses in bold**

Reviewer #1

Manjula-Basavanna, Joshi, and colleagues reported a new process for engineered living materials (ELM) from cellular biomass and engineered curli fibers. Building on their previous work in Aqua plastics, they utilized 1–5% glycerol as a plasticizer to obtain more ductile ELMs in higher yields. The main innovation in this paper is the process of converting cellular biomass and engineered proteins into robust, tangible, and scalable materials comparable to plastics and papers, overcoming the well-known problem of brittleness of dried ELM. I have a few questions and comments for this paper, as noted below. I recommend considering acceptance of this article after the authors could respond to these points fully.

We greatly appreciate Reviewer #1 for their meticulous review and intriguing comments, which has helped us to further improve the clarity of our manuscript. We also thank Reviewer #1 for recommending acceptance of our manuscript.

Congo Red (CR) staining provides a qualitative result and may generate a few different artifacts. According to the SI, the authors treated cell culture with CR solution, removed cells by centrifugation, and measured the absorbance of supernatant at 480 nm. The data was compared with the standard curve prepared with purified CsgA (Figure S9). First of all, I do not understand the rationale for using absorbance values at 410 nm instead of fluorescence, which is more characteristic of the CR in a restricted environment such as amyloid. A previous paper from the Joshi lab (Applied and Environmental Microbiology 2019) describes a higher absorbance reading of CR staining in cell-only samples compared to cell and curli fiber samples (Figures 1 and 2 in the AEM paper), highlighting why fluorescence is more useful for quantifying the amyloid formation. The 410 nm reading will likely give signals from all of the CR molecules in the supernatant, including free-floating CR and non-specifically absorbed CR to the curli fibers (or other biomacromolecules). Moreover, while the standard is prepared with a pure CsgA protein, the staining of CR was done in a more complex mixture, raising questions about whether this is an accurate estimation of the amount of curli fibril. In addition, the standard curve in Figure S9 yielded a linear response between 0.2–0.4 absorbance (corresponding to 0.8–1.2 mg/mL purified protein), but the data from the sample is completely out of that linear range (absorbance around 1.0). We also do not know whether a known number of cells mixed with a known amount of curli fiber would give accurate results with this method. As an alternative to the CR staining, the authors may be able to use immunoblots to estimate the amount of curli

fiber in their system.

We thank the Reviewer #1 for this comment, but there seems to be some misunderstanding, which we would like to clarify. For over a century now, Congo Red (CR) dye has been utilized to stain and provide a qualitative way to identify and visualize the amyloids. However, the quantification of a functional amyloid such as curli has been generally performed with colorimetric CR assays, which involves the measurement of absorbance at 480 nm. In this colorimetric CR assay, the cell culture is incubated with CR, which is then centrifuged to obtain a pellet of the cultured biomass (cells + curli + bound CR). The so obtained supernatant that comprises of unbound CR is used for absorbance (at 480 nm) measurements. We, in our previous publications, and several other groups have routinely utilized this colorimetric CR assay to quantify the curli (Nat. Chem. Biol., Nat. Commun., Nat. Mater., Nat. Nanotech.). For this current manuscript, we went one-step further, to estimate the absolute amount of curli present in the cultured cellular biomass by utilizing a purified CsgA standard curve.

It is to be noted that in one of our previous publications in *Applied and Environmental Microbiology (AEM)*, we had developed an assay for *in situ* real-time monitoring and quantification of curli production in the cell culture. We believe that the Reviewer #1's statement "higher absorbance reading of CR staining in cell-only samples compared to cell and curli fiber samples (Figures 1 and 2 in the AEM paper), highlighting why fluorescence is more useful for quantifying the amyloid formation." is somewhat misplaced due to the following reasons, namely, 1) the recorded absorbance in AEM article was at 600 nm and not at 480 nm, 2) the 600 nm wavelength is also commonly used to monitor the cell density, which essentially shows the scattering of light caused by the cells. This non-linear scattering (non-zero absorbance, typical for colloidal suspensions) across the entire spectrum spanning from 300 to 700 nm is clearly evident in the Fig 2c of AEM article, and 3) CR is relatively a weak fluorescent molecule and thus offers limited signals. As the focus of our AEM article was *in situ* real-time monitoring of cell cultures, which are inherently colloidal in nature, CR fluorescence assay was a better choice than the absorbance-based colorimetric assays to 1) minimize the scattering based artifacts, 2) continuously monitor curli production, and 3) avoid the multiple centrifugation steps to pelletize the cell culture that are necessary for the colorimetric CR assays. However, for this current manuscript, we performed the quantification of curli after culturing the cells for 48 h and there was no need for a real time *in situ* measurements and thus CR fluorescence assay was neither appropriate nor necessary.

To account for the non-specific binding of CR to biomolecules other than curli, we utilize a control of cell pellet having a sham plasmid, wherein the curli operon is deleted (Fig S9a, red bar). In our colorimetric CR assay, we subtract the CR absorbance value of cell pellet having a sham plasmid (FigS9a, red bar) from that of CR dye (FigS9a, pink bar), and this difference (~0.3 a.u.) in CR absorbance value can be considered as the non-specific binding of CR to all other

biomolecules. Further, the difference (~0.5 a.u.) in CR absorbance values of orange and pink bars in FigS9a represents the binding of CR to cell pellet + curli. Finally, by subtracting the net cell pellet absorbance value of ~0.3 a.u from the net cell pellet + curli absorbance value of ~0.5 a.u., we could quantify the net curli amount (~0.2 a.u.) in the cell pellet. Thus, the resulting absorbance difference values of curli, which is ~0.2 a.u., is well within the absorbance range of the purified CsgA standard curve. Therefore, it is evident that our colorimetric CR assay provides a reliable method to estimate the curli amount in the cultured cellular biomass.

Immunoblot analysis of curli in a cellular biomass would be quite complex and tedious, and based on our prior experience, the CsgA antibody shows limited specificity for CsgA (131 AA) fused with relatively large protein domains such as spacer (225 AA), SpyCatcher (113 AA) etc. As the primary objective of this work is on the design and development of a new biofabrication strategy to controllably tailor the mechanical properties of ELMs, we believe that the estimation of curli (which is also consistent with the weight analysis) by an established method like colorimetric CR assay is sufficient and immunoblotting is not necessary and beyond the scope of this manuscript.

The authors also mentioned that the amount of curli fiber species from 500 mL cultures is estimated to be 400–500 mg. Since their estimation method compares the CR staining to the standard solution prepared with a known dry weight of protein (because there is no way to measure the wet weight of protein...), these numbers are the dry weights of individual curli fibers. Also, the wet weight of the pellet obtained in 500 mL culture is ~2500 mg. For *E. coli*, the wet weight is around 1.7 g/L, whereas the dry weight is around 0.39 g/L (23% of the wet weight according to Bionumbers, somewhat less than this but also supported by their dry weight estimation in Figure S10). If their claim is true, their cell culture contains 500 mg of curli protein in 575 mg of cellular biomass (23% of the wet weight, likely less considering their Figure S10), leaving only ~75 mg of everything else in cells. They mentioned that the wet weight of the control (sham plasmid) is ~1900 mg, which corresponds to approximately 437 mg of cellular biomass in dry weight. It does not add up. This is one of the reasons why I think their estimation of curli fiber is significantly off.

We thank Reviewer #1 for meticulously reviewing these finer details, which made us to realize that we should have specified these estimated weights of curli are the wet weights and not the dry weights. His-tag affinity purification was utilized to obtain the pure CsgA proteins. The eluted CsgA protein present in the elution buffer (0.5 M imidazole in phosphate buffer) was buffer exchanged to water, which resulted in protein fibers, that were then pelletized to get the wet weight of pure CsgA. We used this wet weight of CsgA protein based fibers for our standard curve. We have now included this description in the method section and specified the wet weights, wherever we had missed to mention in the earlier manuscript.

Like we have clarified above, the estimated 400-500 mg of curli from 500 mL cultures is of the wet weight and not the dry weight. We apologize for causing this confusion, which we have now corrected in the revised manuscript. Thus, the percentage weight of estimated curli nanofibers to wet cell pellet turns out to be about 20% (Figure 3h).

On the other hand, the percentage of dry to wet cell pellet weight is about 20% (Fig S10), which is close to that obtained by Bionumbers (23%). So, the dry weight of entire cell pellet (wet weight ~2500 mg) would be ~500 mg and ~20% of this meaning ~100 mg is the estimated dry weight of curli. Moreover, the dry weight of sham plasmid cell pellet (20% of 1900 mg) would be 380 mg and thus ~100 mg of estimated dry weight of curli seems reasonable.

Thus, we strongly believe that our rigorous weight analysis indeed provides a satisfactory estimation of curli production.

They quantified the amount of SDS in the gelatinous biomass by EDAX and compared the dry weights of the gel and supernatant. It looks like a significant portion of the biomass is being lost to the supernatant during this process, leaving only 45% of dry weight (whatever is left plus SDS). They took the amount of sodium and sulfur to calculate the amount of SDS in the gel. I don't think this method makes a lot of sense when a significant portion of biomolecules/cellular mass is being lost to the supernatant during this step. Sodium and sulfur elements also exist in the original biomass and cannot give characteristic contrast to enable accurate estimation of SDS.

We agree with the Reviewer #1 that a significant portion of the biomass gets lost into the supernatant. To find out, how much SDS is retained in the MECHS, we utilized EDAX. As rightly pointed out by the Reviewer, sodium and sulfur elements exist in the original biomass and to address this, we had performed EDAX analysis of the cell pellet that was not treated with SDS. This analysis of cell pellet (original biomass without SDS treatment) showed $0.6 \pm 0.1\%$ and $1.2 \pm 0.5\%$ weight percentages of sodium and sulfur respectively. The 3% SDS-treated MECHS showed $2.2 \pm 0.2\%$ and $4.5 \pm 0.3\%$, weight percentages of sodium and sulfur, respectively. Further, we could estimate that 1.6% of sodium and 3.3% of sulfur might have been retained in the MECHS. Thus, we estimate that SDS might contribute to ~5% ($1.6 + 3.3 = 4.9\%$) of the net weight of MECHS (3% SDS-treated). Some of the above finer details are now incorporated into the manuscript text as clarification.

It appears that the amounts of curli proteins expressed (by weight) are similar among CsgA, CL1, and CL2. Curli proteins in CL1 and CL2 are 40–120% and 200-280% longer (and heavier) than CsgA proteins. Assuming the similar expression level of these proteins, the effect of SpyTag-SpyCatcher crosslinking will be negated by the less

amount of CsgA units and decreased amount of assembly. If the weights of MECHS films of CsgA, CL1, and CL2 were in the same range (as shown in Figure S16), why do you see the increase in Young's modulus at all? Are there any differences in expression levels of the curli fiber in these systems?

As rightly pointed out by the reviewer, the fused domains in CL1 and CL2 can account for ~80% (average of 40% and 120%) and ~240% (average of 200% and 280%) of the estimated curli protein length (let's assume it as weights), respectively. Although, we have not verified the expression levels experimentally, it is important to note that the estimated amounts of curli proteins and the wet cell pellet weights of CsgA, CL1 and CL2 are similar. Based on the above details, if we normalize all the CsgA building blocks that form the curli nanofibers in CsgA-MECHS as 100%, then for CL1 and CL2, there could be only 56% and 30% of CsgA building blocks, while the rest of the weight could be contributed by the fused protein domains. In other words, CL1 and CL2 could have only 56% and 30% of the curli nanofibers in comparison to CsgA-MECHS. Despite the significantly lesser curli nanofibers, the observed increase in Young's Modulus of CL1 and CL2 strongly indicates the effectiveness of our crosslinking design strategy involving Spytag and Spycatcher domains. Young's modulus is a measure of material's ability to resist deformation. In the case of CsgA-MECHS, the inter-fiber (neighboring curli nanofibers) interactions are supramolecular in nature. But, in case of CL1 and CL2, the inter-fiber interactions are strengthened by the stronger covalent crosslinking of Spytag and Spycatcher domains and thereby contribute to enhancing the resistance to deformation and thus their Young's modulus increases. The reviewer's comment also presupposes that we can predict precisely how the mass fraction of CsgA (relative to other proteins) in the various constructs affects Young's Modulus, which is not necessarily the case. The presence of linker, SpyTag/Catcher, and unstructured domains may also contribute to mechanical properties. Therefore, we think the crosslinking-based interpretation of the data makes more sense.

How many cells in a given weight or volume are present after SDS and glycerol treatment? I am also curious to know why this material is transparent when (some of the) cells are still intact within the materials. A more detailed investigation of gels with SDS and glycerol (rheology, SEM) before drying can provide more insights into compositions and crosslinking.

As shown in Fig S1 and S3, the SDS and glycerol treated cell pellet is a mucous-like gelatinous biomass. In our previous work, we had investigated the rheological properties of curli hydrogels (both living and non-living), which showed that they are very weak gels with a storage modulus of ~100 Pa. Since the focus of this current work is on plastic-like films, we believe that rheological studies are beyond the scope of this work.

Characterization of this gelatinous biomass, as such by SEM was not favorable due to two reasons. 1) This gelatinous biomass requires chemical fixation, water-washes, solvent exchange steps and critical point drying to preserve and visualize their 3D architectures in SEM. These chemical fixation steps can significantly remove SDS and glycerol from the biomass and thus it will not provide an accurate representation of the composition. 2) Glycerol being a hygroscopic liquid, if present in the biomass, then upon freeze-drying or critical point drying, it would typically coat the cells and nanofibers to form an amorphous spongy/porous architecture and thus inhibits their clear visualization.

Crosslinking of SpyTag-SpyCatcher, which we have utilized to crosslink curli nanofibers, is well-characterized and provided clear evidence through tensile studies that CL1 and CL2 exhibit higher stiffness values. On the other hand, we have performed rigorous weight analysis and EDAX analysis to provide rough estimations of the composition of MECHS.

During the MECHS biofabrication, most of the SDS ends up in the supernatant, which when dried resulted in a brown-yellowish color pellet. Thus, we believe that SDS being a surfactant removes the brown-yellowish color of the cell pellet, which makes MECHS film transparent.

Why does adding plasticizers change the mechanical properties of materials? More molecular explanation and systematic investigation of other plasticizing compounds will provide clues as to why the resulting MECH properties.

Curli nanofibers are assembled from CsgA protein building blocks that comprises of a rigid beta-helical structure, which in simple terms can be regarded as a quasi-crystalline ordering. So, when these curli nanofibers (without plasticizer) based rigid materials are subjected to tensile stress beyond its yield point, the strain induced crack, propagates and it quickly breaks the material. But by incorporating a plasticizer like glycerol, the amorphous nature of plasticizer that surrounds the rigid curli nanofibers inhibit the crack propagation by subjecting the material to undergo plastic deformation. Thus, with increasing plasticizer amounts from 1-5%, the elongation at break was found to increase from 1 to 160%. This description is now included in the revised manuscript. We will employ different plasticizers in our future work to shed greater insights into the correlations of molecular interactions and mechanical properties.

It is unclear how PVA-Mc and PVA-Sp are different.

PVA-Mc and PVA-Sp are two different commercially available PVA products. PVA-Sp (sold by Superpunch) is a water soluble stabilizer for embroidery. It is claimed to be made from 100% PVA and thus it dissolves completely in water. PVA-Mc (sold by Mckesson) is a hot water soluble bag, which is also claimed to be made

from PVA, but it does not dissolve completely in water at room temperature, probably because PVA might be blended with other components. This description is now included in the revised manuscript.

Reviewer #2

In this study, the authors reported Mechanically Engineered Living Material with Compostability, Healability, and Scalability (MECHS) in the form of a stretchable plastic, with the feature of biodegradability, manufacturability, sustainability, and ability to tailor functional properties. The MECHS is produced directly from cultured bacterial biomass, and the fabrication protocol described in this paper increased the yield of bioplastic significantly. The highlight of the work relies on the mechanical properties and biodegradability. Overall, I feel that it could merit publication after minor revision and some necessary revision.

We thank Reviewer #2 for their valuable comments and the recommendation to publish our manuscript.

Major issues:

(1) The authors only mentioned “tried analogous experiments with a large spacer (disordered protein domain of 225 amino acids) in between CsgA and the SpyTag/SpyCatcher domains” on Page 4. Further discussion to explain the reason for inserting IDP between CsgA and the SpyTag/SpyCatcher domains in more detail is encouraged.

We introduced the large spacer for two reasons. 1) To verify if the observed increase in stiffness of CL1 was due to the covalent crosslinking of Spytag and Spycatcher. 2) To test if an intrinsically disordered large protein can modulate the mechanical properties such as stiffness, toughness and elongation at break. We have now included these details in the revised manuscript.

(2) The authors mentioned, “Bigger spacer domain might lead to significant reductions to stiffness and enhanced extensibility.” Have the authors tried proteins with large well-folded structures as Spacers, and what are the expected properties?

In our earlier publications, we have genetically fused CsgA with proteins having specific folded structures (e.g., trefoil factor, fibrin-derived “alpha”/“gamma” protein domains, etc.), which showed significant changes to their mechanical properties, namely, storage modulus. However, we have not tested large well-folded structures as spacers, which we will explore in our future studies. The expected mechanical properties depend on the folded structures, but in general, in our opinion, well-folded structures with high beta-sheet or alpha helix content

might make the material stiffer. This is in line with naturally occurring protein-based biomaterials with low stiffness and high extensibility (e.g., elastin, resilin), which exhibit disordered secondary structure.

(3) The authors mentioned the process of biofabrication of MECHS in SI where the 48 h cell culture was centrifuged, washed, and subsequently added SDS, etc. Further discussion about the comparisons of this approach to other manufacturing methods would be encouraged (ex. the operation difficulty, time, and yield).

In the revised manuscript, we have now included a few additional details (page 8 third paragraph, page 9 second paragraph) on the biofabrication approach of MECHS in comparison to other methods.

(4) The authors demonstrated the MECHS films were able to biodegrade completely in 75 days in fishnure, showing completely compostable features. I am curious about the durability of this material in its use scenarios, would it degrade on its own in use scenarios? Please comment on this.

MECHS do not degrade on their own and the prototypes that were left on our benchtops remained intact even 1-2 years after fabrication. We think that a more detailed investigation of other environmental effects that could affect their function (UV exposure, humidity) is outside the scope of this paper.

Minor issues:

(1) The left part of Figure 1 is not clear enough. It seems like three different kinds of bacteria are co-cultured in the same culture flask. The authors are suggested to modify it to avoid misunderstanding. SpyTag's sequence is much shorter than SpyCatcher's, and the legend of the unfolded peptide should be modified to the actual length.

We have now modified Figure 1 as per the suggestions of Reviewer #2.

(2) Please make sure that the upper and lower case in the title of Materials and Methods is uniform.

We have now corrected the text as per the suggestions of Reviewer #2.

Reviewer #3

The manuscript by Duraj-Thatte and Manjula-Basavanna describes a new E.coli-based ELM called MECHS that consists of dried E.coli cell mass, plasticized with small amounts of glycerol and formed into transparent flexible films that biodegrade readily in compost. Through genetic engineering of the E.coli to produce curli fibres in abundance, the team improve and tune key mechanical properties of their materials. They can also tune properties by changing the glycerol content and use of SDS in turning the cell pellet into a material that can become a film. Finally, the coolest bit of work of all, is having 2 E.coli strains mixed together that produce engineered curli fibres that either have spytag or spycatcher proteins all along them. This means when the different bacteria meet in the biomass their fibres click together, increasing the material strength. The paper has lots of nice materials testing, some very convincing biodegradation experiments and even makes a film to wrap a dishwasher tablet. Cool stuff.

Overall, I really liked this paper and thought it was very well written and presented. I'm not sure if it's been reviewed at another journal beforehand, but it felt like this was a version that had been read, edited and double-checked several times before. I struggled to find anything to correct.

In principle I'm happy for this to be published as it is - it's very good work and good quality.

We thank Reviewer #3 for their generous remarks and the recommendation to publish our manuscript as it is.

My only three comments would be:

1. What is the effect of the culture media composition on the material? i.e. if the E.coli grow in LB in a shake flask, do they make a different quality/strength of material than if they grow on a mixed waste source in a fermentor, instead?

In general, the culture media can influence the growth of E. coli and the production of curli nanofibers. But, if the E. coli is tolerant and/or engineered to proliferate and produce curli on the mixed waste source, then we should be able to obtain MECHS with the same properties. However, the cell density in a fermenter would be several folds higher than that of shake flask and thus we may have to optimize the fabrication process.

2. What is the amount of E.coli liquid culture needed to grow to make 10 cm sq and 1 metre sq? Is it feasible to make a roll of film (e.g 2m long, 5 cm wide) similar to sellotape just by growing a few flasks of bacteria? It would be good to get an idea of scale of production needed to make some amounts that would match real-world use cases.

The MECHS prototype reported in this manuscript of 50 cm × 5 cm lateral dimension was fabricated from 6 L culture of CsgA (obtained by using 3% SDS and 3% glycerol treatment). Therefore, we can get ~40 cm² of MECHS film per liter of culture. So, for 10 cm² and 1 m² of MECHS film, we would require 0.25 L and 250 L of culture, respectively. Yes, it is indeed feasible to make a roll of film with lateral dimensions 2 m × 5 cm and for which we would require 25 L of cultures.

3. The entire two page section 'composition and morphological analysis' was difficult to get through and rather boring. Could the authors consider shortening it or revising it to get to the key points more quickly? I felt like it took forever to get to any key messages from this part of the study.

We believe that the finer details described in the “composition and morphological analysis” section is extremely crucial to understand the effect of various biofabrication steps on the composition of MECHS. We have placed almost all this experimental data in SI, but because of its complex composition, we believe that a detailed description is necessary. To the best of our knowledge, there are no such thorough studies, especially in the field of ELMs and thus it might be instrumental for future studies. We believe that Reviewer #1 was able to raise several intriguing questions, which invariably necessitates the importance of this section to be in the main manuscript. However, as per the suggestion of Reviewer #3, we have made a few modifications to the “composition and morphological analysis” section.

We have addressed all the comments of three reviewers and made necessary modifications to the revised manuscript as per their recommendations. The modified text is highlighted in red. With these changes, we hope that the reviewers recommend acceptance of this manuscript for publication in Nature Communications.

REVIEWERS' COMMENTS

Reviewer #1 (Remarks to the Author):

[Note from the Editor: this reviewer also considered the responses to reviewer 2]

The authors fully explained and addressed my comments. I recommend publishing this study.

Reviewer #3 (Remarks to the Author):

I was already supportive of immediate publication in my first review and only had minor comments. The authors have addressed these in the revision as well as a few technical queries from the other reviewers. Overall, the manuscript has been nicely improved and I'm confident that this is now ready for publication. Well done to the authors.